# Reconciling drainage and receiving basin signatures of the Godavari River system

Muhammed O. Usman[1], Frédérique M.S.A. Kirkels[2], Huub M. Zwart[2], Sayak Basu[3], Camilo Ponton[4], Thomas M. Blattmann[1], Michael Ploetze[5], Negar Haghipour[1,6], Cameron McIntyre[1,6,7], Francien Peterse[2], Maarten Lupker[1], Liviu Giosan[8], Timothy I. Eglinton[1].

[1]Geological Institute, ETH Zürich, Sonneggstrasse 5, 8092 Zürich, Switzerland

[2]Department of Earth Sciences, Utrecht University, Heidelberglaan 2, 3584 CS Utrecht, Netherlands

[3]Department of Earth Sciences, Indian Institute of Science Education and Research Kolkata, 741246 Mohanpur, West Bengal, India

[4]Division of Geological and Planetary Science, California Institute of Technology, 1200 East California Boulevard, Pasadena, 91125 California, USA

[5]Institute for Geotechnical Engineering, ETH Zürich, Stefano-Franscini-Platz 3, 8093 Zürich, Switzerland

[6]Laboratory of Ion Beam Physics, ETH Zürich, Otto-Stern-Weg 5, 8093 Zürich, Switzerland

[7]Scottish Universities Environmental Research Centre AMS Laboratory, Rankine Avenue, East Kilbride, G75 0QF Glasgow, Scotland

[8]Geology and Geophysics Department, Woods Hole Oceanographic Institution, 86 Water Street, Woods Hole, 02543 Massachusetts, USA

*Correspondence to: Muhammed O. Usman (muhammed.usman@erdw.ethz.ch)*

**Abstract.** The modern-day Godavari River transports large amounts of sediment (170 Tg per year) and terrestrial organic carbon ($OC_{terr}$; 1.5 Tg per year) from peninsular India to the Bay of Bengal. The flux and nature of $OC_{terr}$ is considered to have varied in response to past climate and human forcing. In order to delineate the provenance and nature of organic matter (OM) exported by the fluvial system and establish links to sedimentary records accumulating on its adjacent continental margin, the stable and radiogenic isotopic composition of bulk OC, abundance and distribution of long-chain fatty acids (LCFA), sedimentological properties (e.g. grain size, mineral surface area etc.) of fluvial (riverbed and riverbank) sediments and soils from the Godavari basin were analysed and these characteristics were compared to those of a sediment core retrieved from the continental slope depocenter. Results show that river sediments from the upper catchment exhibit higher total organic carbon (TOC) contents than those from the lower part of the basin. The general relationship between TOC and sedimentological parameters (i.e., mineral surface area and grain size) of the sediments suggests that sediment mineralogy, largely driven by provenance, plays an important role in the stabilization of OM during transport along the river axis, and in preservation of OM exported by the Godavari to the Bay of Bengal. The stable carbon isotopic ($\delta^{13}C$) characteristics of river sediments and soils indicate that the upper mainstream and its tributaries drain catchments exhibiting more $^{13}C$ enriched carbon than the lower stream resulting from the regional vegetation gradient and/or net balance between the upper ($C_4$-dominated plants) and lower ($C_3$-dominated plants) catchments. The radiocarbon contents of organic carbon ($\Delta^{14}C_{OC}$) in deep soils and eroding riverbanks suggests these are likely sources of "old" or pre-aged carbon to the Godavari River that increasingly dominates the late Holocene portion of the offshore sedimentary record. While changes in water flow and sediment transport

resulting from recent dam construction have drastically impacted the flux, loci, and composition of OC exported from the modern Godavari basin, complicating reconciliation of modern-day river basin geochemistry with that recorded in continental margin sediments, such investigations provide important insights into climatic and anthropogenic controls on OC cycling and burial.

## 1. Introduction

Rivers form a key component of the global carbon cycle, transporting about 200-400 Tg of particulate organic carbon (POC) to the oceans annually (Degens et al., 1991; Ludwig et al., 1996; Schlünz and Schneider, 2000), with the majority of this POC deposited on the continental margins (Berner 1989; Hedges 1992). Much of this POC is mobilized from soils (Meybeck, 1982; Tao et al., 2015), augmented by recently biosynthesized higher plant debris, recycled fossil OC derived from erosion of sedimentary rocks, and *in situ* aquatic productivity within the rivers (Hedges et al., 1986). Rivers not only act as conduits linking terrestrial and marine reservoirs but also as reactors where terrestrial OC ($OC_{terr}$) is subject to a myriad of processes resulting in degradation and modification of the suspended OC load (Aufdenkampe et al., 2011; Wu et al., 2007; Cole et al., 2007). Although a general framework for describing the origin and evolution of $OC_{terr}$ in different types of river basins is emerging (e.g., Blair and Aller, 2012), a detailed understanding of the impact of the diverse and complex array of processes occurring within river basins on the amount and composition of $OC_{terr}$ that is ultimately exported offshore is still developing.

The flux and nature of OC discharged to the ocean is dependent on a number of factors including the composition of underlying bedrock, geomorphologic properties, as well as climatic factors like temperature and precipitation (Hilton et al., 2008;

Leithold et al., 2006). Climate variability on millennial and longer timescales is considered to exert an important influence on the export of OC from the terrestrial biosphere and burial in ocean sediments, with important feedbacks on atmospheric $CO_2$. Variations in exhumation, oxidation and burial of bedrock OC exported from river basins is considered to exert fundamental controls on atmospheric $CO_2/O_2$ balance over longer (>million-year) timescales (Berner, 2003).

Tropical and subtropical rivers are estimated to account for more than 70% of the global $OC_{terr}$ delivery to the oceans (Ludwig et al., 1996; Schlünz and Schneider, 2000) and thus comprise major vectors in land-ocean carbon transport (Aufdenkampe et al., 2011; Galy and Eglinton, 2011; Hedges et al., 1986; Schefuss et al., 2016; Spencer et al., 2012). The discharge of such rivers is sensitive to variations in climate, such as the location and intensity of monsoonal rains and dry season droughts. Fluvially-derived OC deposited and preserved in adjacent continental margins serve as rich archives of information on past perturbations in continental climate and fluvial dynamics (Bendle et al., 2010; Schefuss et al., 2011; Weijers et al., 2007).

The Godavari River basin (Fig. 1) is an example of a monsoon-influenced low latitude river basin and, as the largest non-Himalayan river in India, is of special interest due to its large catchment size and sediment flux to the ocean (Kale, 2002). Draining central peninsular India, the river integrates rainfall within the core monsoon zone of central India, both reflecting the mean monsoon regime and capturing fluctuations in monsoonal rains over the sub-continent. With over 90% of discharge from the Godavari deriving from summer monsoon precipitation (Rao et al., 2005), corresponding offshore sedimentary sequences record past variations in continental climate as well as anthropogenic activity within the drainage basin (Cui et al., 2017; Giosan et al., 2017; Ponton et al., 2012; Zorzi et al., 2015).

Prior studies of a Godavari River-proximal sediment core NHGP-16A from the Bay of Bengal (BoB) that spans the Holocene have revealed marked geochemical and sedimentological variations that have been interpreted in the context of both evolving regional hydroclimate and accompanying changes in land-use within the Godavari catchment (Giosan et al., 2017; Ponton et al., 2012). Specifically, a distinct change in the stable carbon isotopic ($\delta^{13}C$) composition of molecular markers of terrestrial vegetation implies an increase in the proportion of arid-adapted $C_4$ vegetation beginning around 4.5 ky BP (Ponton et al., 2012). This shift in vegetation type is accompanied by increased variability in the oxygen isotopic composition of planktonic foraminiferal carbonate, suggesting enhanced hydrological variability, potentially reflecting less frequent ("break monsoon"), but intense rainfall activity within the drainage basin (Ponton et al., 2012). Subsequent down-core geochemical and sedimentological measurements on the Godavari-proximal BoB sediments have served to paint a more comprehensive picture of past changes within the Godavari basin (Giosan et al., 2017). Notably, sharp increases in sediment accumulation rate during the late Holocene imply a concomitant increase in fluvial sediment discharge, despite the onset of increasingly arid conditions. Furthermore, detrital Nd isotopic compositions indicate a shift in sediment provenance at ca 4.5 ky BP from a relatively unradiogenic signature consistent with lower basin bedrock as the primary detrital mineral sources prior to increased contributions from the more radiogenic rocks in the upper basin (Deccan plateau). Finally, these changes recorded in the sediment core are also associated with increased [14]C age offsets between bulk OC and coeval planktonic foraminifera, suggesting enhanced erosion and export of pre-aged $OC_{terr}$ exhumed from deeper soil layers (Giosan et al., 2017). Collectively, these different lines of evidence are consistent with an overall scenario in which increasing aridity

results in a shift in the type (from deciduous to shrub or grass) and extent (reduced) of vegetation coverage, while changes in the pattern and frequency of seasonal monsoons promote enhanced soil erosion in the driest regions of the upper basin. Increased soil loss may have been exacerbated by human activity through intensification of agriculture and implementation of irrigation practices that amplify soil disturbance and destabilization (Giosan et al., 2017).

Although the above interpretations appear consistent with available geochemical observations, support is lacking from direct observations on spatial variations in geochemical, mineralogical, and sedimentological properties within the Godavari drainage basin. Furthermore, direct attribution of signatures observed in the sediment record with those of the drainage basin remains elusive. In the present study, we assess the extent to which terrestrial signatures recorded in river-proximal continental margin sediments can be reconciled with those within (specific regions of) the river basin. In particular, we seek to establish whether OC characteristics of the basin are consistent with those of distal sediments deposited during the Holocene on the adjacent continental margin. In addition to bulk and molecular characteristics of particulate organic matter, we explore quantitative and compositional relationships to mineral phases in soils, river and marine sediments. Such an approach comparing between drainage basin and adjacent continental margin signatures may prove crucial in delineating the nature and provenance of signals preserved in marine sedimentary sequences in the receiving basin, and hence for informed interpretation of corresponding down-core records. Specific questions include: (i) To what extents do offshore sedimentary signatures reflect characteristics of the modern-day basin, and what is their provenance? (ii) How and to what degree are organic and mineral matter (de-)coupled during mobilization and transfer from source to sink? By addressing

these questions, we aim to improve our understanding of carbon flow through river basins, as well as to better inform interpretation of geochemical signals preserved in river-dominated sedimentary sequences.

## 2. Materials and methods

### 2.1. Study area

The Godavari River is the largest monsoon-fed river basin and the third largest river (behind Ganges and Brahmaputra) of India, delivering 170 Tg per year of sediment and 1.5 Tg per year of OC to the BoB (Biksham and Subramanian, 1988; Ludwig et al., 1996). It originates from Sahyadris in the Western Ghats and flows toward the east-southeast across the Indian peninsula, traversing various geological and vegetation gradients before emptying into the BoB (Fig. 1). Four major tributaries (Purna, Pranhita, Indravati, and Sabri) drain over 60% of the basin area, and the modern-day catchment ($\sim 3 \times 10^5$ km$^2$) supports a population of about 75 million people (Pradhan et al., 2014). The basin experiences pronounced seasonality with marked wet and dry seasons and the majority of annual rainfall occurs during June-September associated with the moist southwest monsoon winds. The Western Ghats act as an orographic barrier (Fig. 1a and b), strongly affecting the precipitation pattern over peninsular India, with monsoonal rains falling preferentially between the coast and the Ghats, leaving much of the inland region with lower precipitation (Gunnel et al., 2007). As a result, the upper river catchment, spanning the Deccan Plateau, is characterized by arid/semi-arid vegetation and lower annual precipitation (<800-1200 mm/yr), while moist/deciduous vegetation and higher annual precipitation (1600-3200 mm/yr) typifies the lower basin (Asouti and Fuller, 2008) (Fig. 1c i and ii).

The underlying rock formations exert a significant control on sediment and solute

transport by rivers. Based on their erodibility, rock formations in the Godavari are

categorized as follows (Biksham and Subramanian, 1988): (a) Deccan Traps, that are

volcanic in origin and of Tertiary age, are known for their distinct spheroidal weathering

and high fluvial erosion (Subramanian, 1981). The whole Deccan Plateau

(representing $\sim$ 48% of the basin area; Fig. 1) is covered by 10-40 cm thick black clay

loam, which serves as a source of riverine sediments; (b) Sedimentary rocks (mostly

sandstones) of Mesozoic-Cenozoic age located in the central and lower part of the

catchment ($\sim$ 11% of basin area) are known for their high degree of erodibility; (c)

Precambrian granites, charnockites, and similar hard rocks ($\sim$ 39% of total basin area)

are characterized by low erodibility. River tributaries draining through these relatively

stable rock formations (e.g. Sabri and Indravati) carry low sediment loads. Compared

to the Deccan volcanic rocks, soils derived from the erosion of sedimentary and

Precambrian rocks prevalent in the eastern segment of the basin are generally thinner

(<15 cm) and reddish/yellowish in colour (Bhattacharyya et al., 2013). Sediments

transported by the Godavari are thus mostly derived from the Deccan Traps and from

granitoids of the Indian Craton (Biksham and Subramanian, 1988). These contrasting

bedrocks manifest themselves in corresponding isotopic signatures, where relatively

young Deccan volcanic rocks are characterized by highly radiogenic mantle-derived

material ($\varepsilon$Nd = -1±5, $^{87}$Sr/$^{86}$Sr =0.701), while the relatively old Indian Craton is

unradiogenic ($\varepsilon$Nd = -35±8, $^{87}$Sr/$^{86}$Sr =0.716) (Giosan et al., 2017; Tripathy et al.,

2011) (Fig. 1c iii).

Spatial variations in soil types and coverage of the basin are described by (Gupta et

al., 1997). Black soils (vertisols, vertic-inceptisols, and entisols) are prevalent in the

central and western parts of the basin. The eastern part of the basin is dominated by

red/yellow soils (Alfisols and luvisols), and in the estuarine/deltaic region, soil type

varies over relatively short distances (Gupta et al., 1997) (Fig 1c iv).

The Godavari River emerges from the Eastern Ghats on the coastal plain near

Rajahmundry, from where it has built a large delta in conjunction with the neighbouring

Krishna River that empties into the BoB and delivers sediment to the pericratonic

Krishna-Godavari Basin (Manmohan et al., 2003). The latter, located in the central

part of the eastern continental margin of peninsular India, formed as a result of the

downwarping of the eastern segment of the Indian Shield subsequent to the break-up

of Gondwanaland (Murthy et al., 1995). Unlike the Himalayan rivers that adjust to

large-magnitude monsoon floods by increasing their width and width-depth ratio

(Coleman, 1969), the incised channel of the Godavari responds to the increase in

discharge by decreasing its width-depth ratio (Kale, 2002). Because of cohesive banks

and incised channel morphology in the lower basin, shifts in channel position are rare,

resulting in limited overbank sediment deposition and restricted areal extent of the

floodplain. As a consequence of this limited accommodation space in the lower basin,

fluvial sediments either accumulate in the delta or are exported to the BoB.

Furthermore, sediment trapping on the continental shelf is minimal because the shelf

in front of the Godavari Delta is narrow (generally <10 km), promoting more rapid and

direct transport of fluvial sediments to the continental slope. Also, satellite images

reveal a plume of suspended river sediments from the Godavari mouth out into the

BoB past the continental shelf, confirming delivery of riverine sediments to the slope

(Sridhar et al., 2008). Therefore, no major lags in or modifications to the fluvial signals

between discharge from the river and deposition on the continental slope are

expected. This sedimentary regime of the Godavari system thus allows for relatively

straightforward interpretation of sediment sources and transfer processes (Giosan et

al., 2017), and facilitates direct comparison between characteristics of drainage basin and BoB sediments.

Damming of the Godavari River and its tributaries has increased tremendously over the past several decades with more than three hundred hydrologic projects of various sizes in current operation that regulate water discharge and sediment transport to the BoB. For the purpose of our study, we divide the drainage basin into two major sections: the upper basin (UB) section (source to Pranhita River tributary) and lower basin (LB) section (Pranhita to the BoB) as this captures the major contrast in bedrock lithology and vegetation between the two segments, allowing for assessment and attribution of signals emanating from these major parts of the river basin. It should be noted that though the Pranhita River has half of its catchment in the upper basin, about 94% of its total suspended particulate matter (SPM) flux is derived from the Wardha and Wainganga Rivers in the lower reaches of the catchment (Balakrishna and Probst, 2005; Fig. 1) justifying the classification of Pranhita into the lower basin.

**2.2. Sampling**

*2.2.1 River Basin*

River sediments (flood deposits from the flank of the river and riverbed deposits) and soil sampling was carried out in February-March, 2015 coinciding with the dry season. Sampling locations are shown in Fig. 1b, with additional details provided in Table S1. Soil sampling sites were chosen to represent the dominant soil type of the given region, and were sampled on level ground and close to rivers. Surface soils and litter (0-5 cm) were collected using a small hand shovel. Additionally, undisturbed soil profiles were obtained at some targeted locations (Fig. 1b) using a metre-long coring device, and where possible were sampled to bedrock. Soil cores were then sub-

sectioned into a 0-5 cm ("shallow/surface") interval, and every 10cm thereafter ("deep"). These depths were chosen to represent the likely sources of shallow (surface run-off) and deeper (e.g. bank) soil erosion and supply to nearby streams. At a few sites, road constructions provided access to complete soil sections that were sampled at 10 cm intervals.

Riverbed sediments were collected from the middle of the stream either with a Van Veen grab sampler from bridges or with a hand shovel where the river was very shallow. The sampling sites were selected as being representative of the local depositional settings of the rivers and its tributaries, and mostly comprise areas dominated by bedload sediments (channel thalweg) with particles sizes ranging from <2 $\mu$m (clay) to 2 mm (coarse sand) and minor proportions of pebbles and plant debris. Where a tributary joins the mainstem of the Godavari, sampling was conducted before the confluence of the two rivers and shortly downstream of the confluence so as to assess the integrated signal of the sub-catchments.

Where present, riverbank sediments that represent loose and unconsolidated freshly deposited suspended sediments were also collected with a hand shovel and as close to the main river stem as possible. Upon arrival at ETH Zurich, all sediment and soil samples were stored frozen (-20°C), then freeze-dried and subsequently dry-sieved to < 2 mm to remove the rock fragments and plant debris. About 20 soils and sediment sample were further milled to powder using an agate/ball mill.

### 2.2.2. Offshore

A piston offshore sediment core (OS) NGHP-01-16A (16.59331N, 82.68345E, 1268 m water depth) was collected near the mouth of the Godavari River in the BoB (Collett et al., 2014) (Fig. 1b). The 8.5 m-long core spanning the entire Holocene (~ 11kyr; Ponton et al., 2012) was analysed for sedimentological, mineralogical and

geochemical characteristics. Due to the top 25 cm of the core being exhausted by prior investigations, our results are augmented with those from Ponton et al. (2012) and Giosan et al. (2017). The sediment depth corresponding to ~ 4.5 ky BP (ca. 515 cm), representing the onset of the vegetation shift in peninsula India during the Holocene, was designated as the boundary between the early (EH) and late Holocene (LH).

## 2.3. Sample treatment and measurements

### 2.3.1. Mineral surface area

About 1g dry weight (gdw) from each soil and sediment sample (unground) was combusted at 350°C for 6 hours in order to remove the organic matter. The samples were then outgassed at 350°C for 2 hours in a vacuum oven to remove adsorbed moisture on the surface before analysis. Prior to analysis, samples were homogenized in an agate mortar, using a plastic pestle to avoid crushing mineral grains. Surface area of the mineral components of the sediment was analysed by the multi-point BET $N_2$ adsorption method using a Quantachrome Monosorb Analyser (Wakeham et al., 2009). The precision on duplicates of alumina standards was better than 1%.

### 2.3.2. Grain size

An aliquot (~0.5 gdw) of combusted (350°C, 6h) sediment and soil samples processed for mineral surface area analysis was treated with 10-15 mL of dissolved (40 g/L) sodium pyrophosphate ($Na_4P_2O_7.10H_2O$) for about 12 hours to disaggregate the sediment grains. Sediment and soil grain size distributions were measured using a Malvern Mastersizer 2000 Laser Diffraction Particle Analyser that characterizes particle sizes ranging from 0.04 to 2000 μm. Sediment and soil samples were measured in triplicate, with average median (d50) values reported. The standard deviation on triplicate analysis was better than 1%.

### 2.3.3. Sediment mineralogy

Eight sediment and soil samples were selected to represent varying regions and lithologies of the Godavari basin, and 12 samples taken from various depths in offshore core 16A were selected for x-ray diffraction (XRD) analysis. About 1g of bulk sediment was wet-milled in ethanol using a McCrone Micronising mill. The milled sample was then passed through a 20 μm sieve and transferred into a ceramic bowl. Mineral grains larger than 20 μm were reintroduced into the mill and the process was repeated. The milled samples were dried overnight at 65°C. The dried sample was pulverized and homogenized using a Fritsch Pulverisette 23 milling device. The resulting sample was then gently loaded onto a sample holder and packed continuously using razor blades to form a randomly-oriented powdered specimen with a smooth surface which minimizes preferential orientation (Zhang et al., 2003). A second sample preparation was carried out producing textured specimens for enhancement of the basal reflections of layered silicates thereby facilitating their identification. The changes in the basal spacing in the XRD pattern by intercalation of organic compounds (e.g. ethylene glycol) and after heating (2 h, 550°C) were used for identification of smectite and kaolinite, respectively. XRD measurements were performed on a Bruker AXS D8 Theta-Theta diffractometer using Co-Kα radiation. The instrument was equipped with an automatic theta compensating divergence and antiscattering slit, primary and secondary soller slits and a Sol-X solid state detector. The phase composition was the determined using the DIFFRACplus software. Mineral phases were identified on the basis of the peak position and relative intensity in comparison to the PDF-2 database (International Centre for Diffraction Data. Quantification of minerals was achieved with the BGMN/AutoQuan software using Rietveld refinement (Bergmann and Kleeberg, 1998; Bish and Plötze, 2011).

### 2.3.4. Bulk elemental and isotopic analysis

Aliquots of freeze-dried sediment or soil samples (~50-200 mg) were weighed into pre-combusted silver boats (Elementar) and fumigated in a closed desiccator in the presence of 12M HCl (70°C, 72 hours) to remove inorganic carbon (Bao et al., 2016; Komada et al., 2008). The samples were subsequently neutralized and dried over NaOH pellets to remove residual acid. The sample was then wrapped in tinfoil boats (Elementar), pressed, and analysed using a coupled elemental analyser-isotope ratio mass spectrometer-accelerator mass spectrometer (EA-IRMS-AMS) system at ETH Zurich (McIntyre et al., 2016; Wacker et al., 2010). The instrumental set-up, blank assessment, accuracy, and reproducibility for the data presented here have been previously reported in McIntyre et al. (2016).

For down-core sediments, $^{14}C_{OC}$ values were decay-corrected for $^{14}C$ loss since time of deposition (eq.1; Stuiver and Polach, 1977). This decay correction is necessary to facilitate comparison of $^{14}C$ values between the sediment core and $^{14}C$ signatures in the modern river basin. The decay-corrected radiocarbon level, $\Delta$ is calculated as:

$$\Delta = (F^{14}C \; e^{\; \lambda \, (1950-x)} - 1)*1000 \qquad \text{eq. 1}$$

where $F^{14}C$ = measured fraction modern value of $^{14}C$, $\lambda = (\ln2)/5730 \; yr^{-1}$ (5730 years is the true half-life of $^{14}C$), x = year of deposition. The year of sediment deposition is estimated from the age model of Ponton et al. (2012) (Table S2). Henceforth, all bulk $^{14}C$ values for the offshore sediment core refer to the $\Delta$ value. However, it should be noted that the influence of "bomb $^{14}C$", resulting from above-ground nuclear weapons testing in the mid-20[th] century, on modern-day Godavari basin $^{14}C$ values is not accounted for in this calculation

### 2.3.5. Compound-specific stable carbon isotopic analysis

Freeze-dried and homogenized sediment samples (30-100g) were microwave-extracted with dichloromethane (DCM): methanol (MeOH) (9:1 v/v) for 25 min at 100°C (MARS, CEM Corporation). The 20 selected milled samples were extracted, using an Accelerated Solvent Extractor (ASE 350, Dionex, Thermo-Scientific), with DCM:MeOH (9:1 v/v) at 100°C and 7.6 MPa. The total lipid extracts (TLE) were dried under $N_2$ and then saponified with 0.5 M potassium hydroxide (KOH) in MeOH (70°C for 2 h). A "neutral" fraction was obtained by back-extraction with hexane after addition of Milli-Q water with sodium chloride (NaCl) to aid separation. The "acid" fraction was obtained by back-extraction of the hydrolysed solution with hexane:DCM (4:1 v/v) after adjusting the pH to ≤ 2. The acid fraction was transesterified with MeOH:HCl (hydrochloric acid) (95:5 v/v) of known isotopic composition at 70°C for 12-16 h in order to yield corresponding fatty acids methyl esters (FAMEs). The resulting FAMEs were then purified using silica gel-impregnated silver nitrate ($AgNO_3$-$SiO_{2)}$) column chromatography to remove unsaturated homologues. Aliquots of the FAMEs obtained from each sample were measured in duplicate by gas chromatography-isotope ratio mass spectrometry (GC-IRMS) using an HP 6890 GC coupled with a Thermo-Delta V IRMS system. The $^{13}$C values of fatty acids (FAs) were subsequently corrected for the contribution of the added methyl carbon and respective errors were propagated (Tao et al., 2015). The average uncertainty is 0.3‰ for the n-FAs. Results are reported relative to Vienna Pee Dee Belemnite (VPDB) (Craig, 1953).

## 3. Results

### 3.1. Surface and deep soils

Both surface and deep soils from the upper basin are highly enriched in smectite (30-

50% of total minerals) with lesser abundances of kaolinite and illite+chlorite (Fig. 2).

On the other hand, soils from the lower basin are mostly quartzo-feldspathic (25-40%

of total minerals) with minor amount of kaolinite (Fig. 2; see also Kulkarni et al., 2015;

Subramanian, 1981). TOC contents of Godavari River basin soils range from 0.1% to

1.8% (mean= 0.6 ± 0.4 %, $n$=67; Table S1). The highest TOC values were found for

surface soils close to the headwaters of the river (Fig. 3a). The highest and lowest

values for median grain size (GS) (970µm and 5.9µm, respectively) are recorded in

surface soils from the upper part of the basin (Table S1). High mineral surface area

(MSA) values are common in upper basin soils (mean=42 ± 18 $m^2$/g, $n$=51) with lower

values in those from the lower basin (mean=21 ± 11 $m^2$/g, $n$=16; Fig. 3b). MSA-

normalized OC (a term that expresses OC loading on mineral surfaces) values of soils

range from 0.03 – 0.84 mg OC/$m^2$ (mean = 0.20 ± 0.15 mg OC/$m^2$). Due to the

relatively low TOC values and high MSA values, the majority of the soils plot outside

the range of typical river suspended sediments as defined by Blair and Aller (2012).

Lipid analyses from river sediments and soils produced LCFA with an average chain

length (ACL) consistently > 28 and similar stable carbon isotope values among $C_{26}$-

$C_{32}$ FA homologues (Supplementary Figure 1). Thus, isotopic values are reported as

mean weighted averages of $C_{26}$-$C_{32}$ FA (Table S1). LCFA of soils range from 4 to 264

µg/g OC with extremely low concentrations in the surface soils of the lower basin

(mean = 10 ± 3 µg/g OC, $n$=8). SA-normalized $C_{26-32}$ FA concentrations (FA loadings)

of surface soils range from 0.1 – 4.6 µg LCFA /$m^2$ (mean = 2.0 and 0.3 µg LCFA/$m^2$

for upper basin and lower basin soils, respectively) and decrease progressively

towards the estuary. The average $\delta^{13}C_{OC}$ value of upper basin soils (-17.9 ± 3.1 ‰;

$n$=51) contrasts sharply with that of soils from the lower basin (-23.2 ± 2.0 ‰; $n$=16,

Fig. 3c). A similar ~ 5 ‰ difference was observed in corresponding $\delta^{13}C_{LCFA}$ values,

which average -24.1‰ (±0.3‰, $n$=39) in the upper basin and -30.6‰ (±0.3‰, $n$=8) in the lower basin.

The soil depth profiles generally show a decrease in TOC contents from top to bottom accompanied with relatively invariant (upper basin) or increasing (lower basin) $\delta^{13}C_{LCFA}$ values (Supplementary Figure 2). Corresponding $\Delta^{14}C_{OC}$ values of soils range from -337‰ to +132‰ with the most depleted values recorded in deeper soil horizons (Fig. 3d).

### 3.2. Riverbed and riverbank sediments

The median GS of riverbed and riverbank sediments varied between 8–851µm (Table S1). Generally, the upper basin is characterized by fine-grained sediments (9-50 µm; mean = 23 ± 11 µm, $n = 12$), and the lower basin by coarse-grained sediments (136-852 µm; mean = 456 ± 288 µm, $n = 6$) (Table S1), except in the delta where finer grained material (8-116 µm, mean = 51 ± 57 µm, $n = 3$) again predominates. Similarly, MSA values show consistently high values (19-60 $m^2/g$; mean = 39 ± 11 $m^2/g$, $n = 12$) in the upper basin, markedly lower values in the lower basin (2-14 $m^2/g$; mean = 6 ± 4 $m^2/g$, $n = 6$) and intermediate values in the delta (12-37 $m^2/g$; mean = 28 ± 13 $m^2/g$, $n = 3$; Table S1, Fig. 3b). There is a weak positive linear correlation between MSA and GS ($r^2$ = 0.33 and 0.36 for riverbank and riverbed sediment, respectively). Samples with lower GS and higher MSA generally have higher TOC contents (0.3-1.6 %; Fig 3a and 3b). Conversely, sediments with coarser GS/lower MSA have low TOC contents (0.1-0.4%). OC loading values, which range from 0.09 – 0.80 mg OC/$m^2$ (mean = 0.29 ± 0.18 mg OC/$m^2$) are generally low compared to typical river sediments (Fig. 4; Blair and Aller, 2012; Freymond et al., 2018). The relative phyllosilicate mineral content of the two upper basin sediment samples that were analysed shows that

smectite predominates (65% & 72% for riverbed and riverbank, respectively) while the kaolinite content is higher in the riverbed (10% versus 1% for riverbank). The remainder of the phyllosilicates comprises illite and chlorite (25% & 27% for riverbed and riverbank, respectively; Fig. 3).

Concentrations of long-chain ($C_{26-32}$) FA in riverine sediments range from 7-187 µg/g OC, with lowest concentrations (7-16 µg/g OC) found in the lower basin riverbed sediments, and highest concentrations (67-187 µg/g OC) in upper basin riverbank sediments (Table S1). The lower basin riverbed sediments have very low $C_{26-32}$ FA concentrations (mean = 9 ± 4 µg/g OC, $n$=4), while those of upper basin riverbed sediments are below detection. Bulk $\delta^{13}C_{OC}$ values for all river sediments range from -17.3‰ to -25.2‰. The most enriched $\delta^{13}C_{OC}$ value was observed in the upper basin (Fig. 3c), where samples yielded an average value of -20.6‰ (±0.3‰, $n$=12). The most depleted $\delta^{13}C_{OC}$ value was recorded in the lower basin, where fluvial sediments averaged -24.1‰ (±0.3‰, $n$=9). Compound-specific $\delta^{13}C$ analysis of LCFA ($\delta^{13}C_{LCFA}$) of river sediments yielded values between -24.8‰ to -32.8‰. Similar to $\delta^{13}C_{OC}$, the most enriched $\delta^{13}C_{LCFA}$ value (-24.8‰, mean=-27.4±0.3‰, $n$=6) was recorded in the upstream section and the most depleted value (-32.8‰, mean=-31.3±0.3‰, $n$=4) observed in the downstream segment. The $\Delta^{14}C_{OC}$ values of river sediments vary between -151‰ and 97‰ (Fig. 3d) with no clear systematic difference between the upstream and downstream sections of the basin.

### 3.3. River-proximal marine sediments

The TOC content of sediments from core 16A varies from 1.2% to 2.1%, and generally decreases from bottom to the top of the core, resulting in mean values of 1.9 ± 0.1% ($n$=10) and 1.6 ± 0.3% ($n$=37) for the early and late Holocene, respectively (Table S2, Fig. 3a). The GS and MSA are both fairly uniform with values ranging from 4.0 µm to

5.2 μm (mean = 4.5 ± 0.3 μm) and 54 $m^2$/g to 72 $m^2$/g (mean = 63 ± 4 $m^2$/g),

respectively. OC loadings decrease progressively from the early (mean= 0.31 ± 0.03 mg OC/$m^2$) to late Holocene (mean = 0.25 ± 0.04 mg OC/$m^2$). The range OC loading values of the core sediments (0.19 – 0.35 mg OC/$m^2$) are within the range of values expected for deltaic and deep-sea sediments (Fig. 4; Blair and Aller, 2012). Relative abundances of phyllosilicate minerals of the analysed core sediments show that early Holocene sediments have slightly higher kaolinite and illite+chlorite contents than the late Holocene sediments, whereas the smectite contents of late Holocene sediments are slightly higher (Fig. 2).

Concentrations of LCFA in the core vary between 49 and 519 μg/g OC (mean = 181 ± 95 μg/g OC), but remain relatively invariant despite the down-core variations in TOC (Table S2). LCFA loading ranges from 0.82 to 4.93 μg LCFA/$m^2$, with slightly higher loadings in the late Holocene (mean = 2.73 ± μg LCFA/$m^2$) than the early Holocene (mean = 2.15 μg LCFA/$m^2$). These are similar to the range of LCFA loading values observed in the Danube Basin (Freymond et al., 2018). Stable carbon isotopic compositions of bulk OC ($\delta^{13}C_{OC}$) range from -19.9 ‰ to -18.2 ‰ (mean = -18.8 ± 0.5 ‰, $n$ = 35) and -20.8 ‰ to -19.8 ‰ (mean = -20.3 ± 0.5 ‰, $n$ = 8) for the late and early Holocene, respectively. $\delta^{13}C_{LCFA}$ range from -26.75‰ to -23.43‰ (mean = -24.89±1.16‰, $n$ = 34) and -28.90‰ to -26.84‰ (mean = -28.01±0.48‰, $n$ = 11) for the late and early Holocene, respectively. There is a gradual increase in both the $\delta^{13}C_{OC}$ and $\delta^{13}C_{FA}$ values towards the top of the core. The Δ values of the measured samples (corrected for decay since deposition) vary between -194.6 and 52.1‰, and generally increase with increasing depth (Table S2).

**4. Discussion**

### 4.1. Evolution of organic matter-mineral associations in the Godavari River basin

Soil and sediment samples analysed from the Godavari River and its major tributaries reveal a wide range of grain sizes, mineral surface areas, as well as TOC contents and compositions. This diversity in characteristics encompasses the range of values reported in previous studies of river sediments and soils within the Godavari catchment (e.g. Balakrishna and Probst, 2005; Pradhan et al., 2014; Kulkarni et al., 2015; Cui et al., 2017). The average TOC content of the upper basin riverbed sediments is a factor of 2 higher than that of lower basin sediments (0.9±0.5% versus 0.4±0.4%, Table S1), and this distribution reflects the geochemical and sedimentological characteristics of the basin. The relatively high TOC values in the upper basin are likely due to low suspended sediment loads and/or greater proportions of organic debris (Ertel and Hedges, 1985). The modern-day upper Godavari is characterized by low suspended sediment load and relatively high phytoplankton production, resulting in relatively high OC contents in riverbed sediments (Pradhan et al., 2014). In contrast, the lower catchment is more heavily charged with suspended sediments primarily derived from the Pranhita and Indravati Rivers draining the Eastern Ghats (Balakrishna and Probst, 2005), with dilution by lithogenic materials resulting in lower observed TOC values in sediments from the lower Godavari basin. The lithological contrast between the upstream and downstream part of the basin may also play an important role in OM distribution between both parts of the basin (see Fig. 1). Erosion of the basalt in the upper basin produces high MSA, smectite-rich clay mineral assemblages, whereas erosion of granitic rocks outcropping in the lower basin yields lower-MSA, kaolinite-rich assemblages. This lithological contrast likely accounts for the spatial offset in the MSA between the upstream and downstream Godavari (Fig. 2b). The higher TOC

values are likely a result of availability of high mineral surface that provides substrate

for OM sorption, stabilization and protection (Keil et al., 1997; Arnarson and Keil, 2007; Gordon and Goni, 2004; Mayer, 1994b). Organic matter first develops associations with minerals during soil formation (Mayer, 1994a), and these organo-mineral associations that evolve during soil mobilization and erosion are considered to influence the balance of preservation and oxidation (Marin-Spiotta et al., 2014; Wang

et al., 2014).

Plotting mineral surface area normalized OC (OC/MSA) versus $\delta^{13}C_{OC}$ for the river basin sediments and soils reveals marked differences between upper and lower basin signals (Fig. 5a). Relatively low OC loading and higher $\delta^{13}C_{OC}$ values characterize the upper basin whereas high OC loadings and lower $\delta^{13}C_{OC}$ values typify the lower basin.

This likely reflects the spatial contrast in vegetation and sedimentology/mineralogy within the river basin. Higher $\delta^{13}C_{OC}$ values in the upper basin reflect a greater preponderance of $C_4$ vegetation in the upper basin while lower OC loadings are attributed to a wide range of factors including erosion of heavily weathered soils that are relatively depleted in OC, and notably enriched in high-surface area smectite-rich

secondary minerals due to erosion of basalts of the Deccan Plateau (Table S1). This interpretation is consistent with other independent observations within the Godavari Basin and its adjacent margin (Kessarkar et al., 2003; Philips et al., 2014; Shrivastava and Pattanayak, 2002; Srivastava et al., 1998). Assessment of relationships between OC loading and $\Delta^{14}C_{OC}$ show that samples with higher OC loadings are generally more

enriched in [14]C (Fig. 5b). In contrast to bulk OC loadings, LCFA loadings are generally higher in the upper basin (Fig. 6a). These low OC/high LCFA loadings in the upper basin suggest that a large proportion of the OC stabilized onto mineral surfaces derives from terrestrial plants, even at low OC contents. Furthermore, like $\delta^{13}C_{OC}$

values, $\delta^{13}C_{LCFA}$ values are relatively high in the upper basin (Fig 6b), indicating a predominant $C_4$ plant origin.

Coupled plots of $\delta^{13}C$ versus $\Delta^{14}C$ have been widely used to elucidate potential sources of OC in riverine systems, and to delineate various end-member contributions to OC (e.g., Goni et al., 2005; Marwick et al., 2015). The Godavari basin samples exhibit a broad range of $\delta^{13}C_{OC}$ values, indicative of mixed vegetation signatures of savanna, tropical grasslands, and tropical forests, as well as aquatic productivity and bedrock inputs, with higher $\delta^{13}C_{OC}$ and $\delta^{13}C_{LCFA}$ values of upper basin sediments and soils reflecting the greater proportion of $C_4$ (versus $C_3$) vegetation. When plotted in $\delta^{13}C$ versus $\Delta^{14}C$ space (Fig. 7), the majority of the upper basin sediments and soils plot within the "soils" end-member and generally cluster around the $C_4$-plants domain, whereas most of the lower basin sediments and soil plot within the vicinity of the $C_3$-plant end-member. This implies that OC in the upper basin sediments mostly derive from $C_4$ plant-derived soils OM with a minor $C_3$ plants contribution, as evidenced by the clustering of sediments around the $C_4$ end of the soil domain. In the same vein, lower basin samples point to increased contribution of $C_3$ plant-derived terrestrial OM (soil).

The spatial decoupling of upper and lower basin geochemical signatures of river sediments has been largely attributed to the vegetation gradients in the basin. However, the apparent lack of upper basin signatures in fluvial sediment from the lower reaches could also be a consequence of in-river processes such as loss/replacement of OC and/or sediment dilution. The general increase in $\Delta^{14}C$ values from upper to lower basin (Fig. 8) indicates that preferential loss of younger, more reactive fraction is unlikely. Modern sediment and OC flux data show the highest POC yield ($\sim$ 12 t/km$^2$/yr) in the Indravati and Pranhita Rivers mostly as a consequence of

high runoff that carries large amount of (younger) plant detritus and loose (top) soil from the forest to the mainstream (Balakrishna and Probst, 2005). Presently, more than 470 km$^2$/year are lost in the lower basin due to deforestation and forest fire, with maximum forest denudation taking place in the state of Orissa (Silviera, 1993), which is drained by the Indravati River. These processes may destabilize soils and enhance loss of associated OM to the fluvial network. In contrast, the general decrease in TOC contents towards the lower basin (Fig. 3a) and downstream increase in SPM (Gupta et al., 1997) points towards dilution of riverine OC with mineral matter derived from soil erosion in the lower basin. As a result, the OC signatures in the modern-day Godavari river sediments appear to not only reflect the biogeographic and geochemical make-up of the basin, but also the processes (loss and replacement versus sediment dilution) that influence the nature of OC.

### 4.2. Linkages between Godavari drainage basin and marine sedimentary signals

The Holocene record from core 16A (Fig. 5, 7, 8; Table S2) shows that increasing long-chain plant wax $\delta^{13}$C values from early to late Holocene coincide with other lines of evidence indicating a transition to drier conditions on the Indian and Arabian peninsulas (Ponton et al., 2012; Prasad et al., 2014). Because $C_4$ vegetation is adapted to more arid conditions, the marked isotopic change beginning at $\sim$ 4.5 ky BP, accompanied by a shift in neodymium isotopic composition towards Deccan bedrock signatures (Tripathy et al., 2011) in detrital phases, has been interpreted to reflect a shift in sediment provenance associated with changes in basin hydrology, resulting in increased sediment flux from the upper Godavari catchment to the adjacent continental margin (Giosan et al, 2017).

In contrast to the river basin sediments and soils, the uniform distribution of grain size and mineral surface area in receiving basin sediments is likely a result of

hydrodynamic sorting during fluvial transport and export of sediments to the BoB.

Thus, in order to compare and contrast signals emanating from the Godavari drainage basin with those in sediments deposited on the adjacent continental margin, it is important to take into account processes that may induce particle mobilization, transformation and sorting. Normalization to MSA may provide a means to address this problem, as it eliminates hydrodynamic sorting effects due to GS, particle density,

and shape (Freymond et al., 2018). Marked differences between early versus late Holocene offshore sediments that mimic upper versus lower basin signals, respectively, are evident when MSA normalized OC (OC/MSA) is plotted versus $\delta^{13}C_{OC}$ (Fig. 5a and 8). The mean OC loadings of early ($0.33\pm0.03$ mg OC/m$^2$) and late Holocene ($0.25\pm0.04$ mg OC/m$^2$) sediments are similar to mean loading values

observed in lower ($0.31\pm0.12$ mg OC/m$^2$) and upper basin ($0.24\pm0.15$ mg OC/m$^2$) riverbed sediments, respectively. In addition, these values are similar to the OC loadings of soils from the respective source regions in the basin ($0.29\pm0.14$ mg OC/m$^2$ and $0.23\pm0.17$ mg OC/m$^2$ for lower and upper basin soils, respectively). The early Holocene part of the record is characterized by relatively high OC loading and lower

$\delta^{13}C_{OC}$ values that progressively shift towards lower OC loading and relatively higher $\delta^{13}C_{OC}$ values during the latter part of Holocene (Fig. 5).

There have only been limited investigations on the longitudinal evolution of OM-mineral interactions during transit through river basins (Freymond et al., 2018). However, evidence suggests that loss and replacement of OM may be substantial

within floodplains, estuarine and deltaic systems (Galy et al., 2008; Keil et al., 1997). Estimates of MSA in marine sediments are complicated by the production and deposition of biogenic carbonate and opal (Hobert and Wetzel, 1989). However, sediment trap data from the central BoB suggest that modern-day carbonate and opal

fluxes to BoB are relatively low (0.03 – 3.1 g/m$^2$ per year; Sarin et al., 1979). In

addition, low foraminifera abundances and high sedimentation rates supported by

detrital sediment inputs (Giosan et al, 2017), especially during the late Holocene,

minimize the effect of carbonate and opal influences on MSA measurements at this

location. Consequently, the measured MSA was interpreted as exclusively reflecting

fluvially-derived lithogenic materials. In this context, we do not find any systematic

difference in MSA between early and late Holocene sediments, with OC loadings that

plot within the general range that is characteristic of deltaic and deep-sea sediments

(Fig. 4; Blair and Aller, 2012).

Bulk OC loading versus $\delta^{13}C_{OC}$ and $\Delta^{14}C$ show that at higher loading, OC is relatively

$^{13}$C-depleted and enriched in $^{14}$C, whereas the reverse is the case at lower loading

(Fig. 5). Direct comparisons of bulk OC loadings between marine sediment core and

river basin soils/sediments are not straightforward as the likely addition of marine

carbon to offshore sediments introduces a layer of complexity to such comparison. In

contrast, LCFA derive exclusively from terrestrial higher plants enabling more direct

comparison of loadings between riverine and offshore sediment. Adopting the

biomarker loadings concept described by Freymond et al. (2018), we find elevated

LCFA loading in the upper basin compared to the lower basin (Fig. 6a and 8), and a

similar range of LCFA loadings in sediments deposited during the early and late

Holocene to that observed in soils and sediments of the lower and upper basin,

respectively. This suggests that the loading signatures in early versus late Holocene

are likely a consequence of the changes in sediment provenance previously inferred

from neodymium isotopic data (Giosan et al., 2017).

The progressive increase in stable carbon isotopic values of bulk ($\delta^{13}C_{OC}$) and long-

chain fatty acids ($\delta^{13}C_{LCFA}$) from the marine sediment core towards the late Holocene

has been interpreted as enhanced supply of $C_4$-derived OC sourced from the Deccan Plateau during the late Holocene triggered by changes in Indian monsoon strength and/or location (Gadgil et al., 2003; Sinha et al., 2011; Webster et al., 1998). Our new results from within the drainage basin lend support for a significant reorganization in sediment and OC provenance from lower to upper basin sources.

The decay-corrected $^{14}$C values of the sediment core are bracketed by the range of values of surface and deep soils from the upper and lower basin (Fig. 7&8). This suggests that the core consist of a mixture of pre-aged carbon sourced from deep soils and fresh carbon from plant litters and possibly freshwater algae. It should however be noted that soil samples, including deeper soil layers have likely been impacted by "bomb $^{14}$C" (see Trumbore at al., 1989, van der Voort at al., 2017). There is a general decrease in the Δ values towards the late Holocene, and the ranges of Δ values of the late Holocene sediment are only observed in the deeper sections of upper basin soils (Supplementary Fig. 2).

Taken together, these sedimentological and geochemical results suggest that export of OC-poor $C_4$-dominated/smectite-rich mineral soils intensified in the late Holocene, with the observed shift ca 4.5 ky BP reflecting a shift in sediment provenance from lower basin to upper basin. Furthermore, much of these upper basin sediments were likely derived from deeper, older, more degraded Deccan soils. This apparent shift in the loci and nature of soil mobilization is also accompanied by a three-fold increase in sediment flux (Pradhan et al., 2014), implying extensive soil loss from the upper catchment (e.g. Van Oost et al., 2012). This loss may have stemmed from both natural (aridification and associated reduction in vegetation cover) and anthropogenic (agriculture and irrigation) causes, the latter potentially being triggered by changes in regional climate.

For the period spanning the late Holocene, perturbations within river basins due to

natural climate variability have become intertwined with those stemming from human

activity. This is particularly so for subtropical river basins of central Asia where the

influence of anthropogenic activity on the landscape and watersheds extends back

several millennia (e.g., Van Oost et al., 2012). Within the past two centuries, humans

have imparted particularly dramatic changes on drainage basins both in terms of land-

use (e.g., deforestation, agricultural practices) and modification of water networks

through dam construction and other major perturbations (Syvitski et al., 2005).

Both the landscape and hydrological characteristics of the Godavari basin have been

dramatically altered over the past century. For example, in the past few decades, there

has been a tenfold decrease in OC flux from the Godavari to BoB due to reduced

monsoon rainfall and dam constructions (Gupta et al., 1997; Pradhan et al., 2014).

However, the late Holocene section generally mimics modern-day upper basin

signatures in high fidelity, suggesting that the perturbations of the modern Godavari

had little impact on sediment and OC mobilization.

The general agreement between signals emanating from the river basin and those

recorded in the sedimentary archive provides valuable insights into understanding the

major mechanisms of sediment and OC mobilization, the dynamics and interactions

of organic matter and sedimentary minerals during fluvial transport, and their impact

on the provenance and nature of signals exported from the drainage basin.

Furthermore, such studies that seek to reconcile drainage and receiving basin

characteristics, and climate and anthropogenic influences on these connections, are

necessary to determine factor(s) controlling the nature and fate of OC preserved in

sedimentary archives.

**5. Conclusions**

In this study, we sought to reconcile previously observed geochemical variations in the Holocene sediments deposited in the BoB offshore of the Godavari River with those observed in soils and sediments within the modern drainage basin.

- Distinct contrasts were observed in the abundance and characteristics of OM and mineral components of soils and fluvial sediments in the upper and the

lower basin. The former (upper basin) are characterized by $C_4$-dominated OM associated with high surface area Deccan-sourced mineral phases, whereas those of the lower basin contain higher proportions of $C_3$ plant-derived OM.

- The strong links between OM characteristics and sediment mineralogy (GS, MSA) suggest that OM-mineral interactions play an important role in OC

stabilization throughout the Godavari source to sink system, from mobilization to export.

- Comparison of bulk and molecular-level characteristics of drainage basin and marine sediment core show a marked mid-Holocene transition is consistent with a change in sediment provenance towards a greater contribution of

Deccan-sourced material in the upper basin. Although, extensive anthropogenic perturbation of the modern Godavari system limits the effective transmission of upper signal to the deltaic region and offshore. However, given the limited accommodation space that restricts upstream trapping and promotes rapid export, anthropogenic influences on the flux and nature of OC

exported from the Godavari basin may be subject to marked future changes.

- Our findings suggest that reconstruction of past continental conditions based on terrestrial biomarker proxy records in marine sediments need to consider

potential shifts in signal provenance as a consequence of both natural and

anthropogenic forcing


**Acknowledgements**

We thank the associate editor Markus Kienast and two anonymous reviewers for the

comments. This project was supported by the Swiss National Science Foundations

("CAPS LOCK" Grant no: 200021-140850 and "CAPS-LOCK2" Grant no: 200021-

163162). Francien Peterse received funding from NWO-Veni grant (grant no:

863.13.016). Liviu Giosan thanks colleagues and crew from the NGHP-01 expedition

for intellectual interactions leading to pursuing work on fluvial-continetal margin

systems of Peninsular India and to grants from the National Science Foundation

(OCE-0841736) and Woods Hole Oceanographic Institution. We also wish to thank

Daniel Montluçon for laboratory assistance. We also wish to acknowledge the

logistical support of Dr. Prasanta Sanyal and Chris Martes with sampling. Further

thanks to Michael Strupler for help with grain size measurements. This manuscript

benefitted from discussions with Chantal Freymond.

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

**Figure captions:**

Figure 1. (a) Location of the Godavari river basin in central-peninsula India. (b)

Sampling locations along the river basin. Upper (UB) and lower basin (LB) samples

are shown in red and blue colours, respectively (Modified from Pradhan et al., 2014). (c) The Godavari drainage basin in its ecological (i), hydroclimatic (ii), geological (iii), and soil cover (iv) context (Modified from Giosan et al., 2017).

Figure 2. Phyllosilicate mineral composition of the Godavari River basin and offshore
sediments.

Figure 3. (a) Total organic carbon (OC) % (b) Mineral surface area (MSA) ($m^2$/g) (c) Bulk OC stable carbon isotope $\delta^{13}C_{OC}$ (per mil) (d) Bulk OC radiocarbon contents ($\Delta^{14}C_{OC}$) (per mil) in the Godavari basin and adjacent margin. Symbols key and
abbreviations are as for Fig. 1 and 2. Average values of early (EH) and late Holocene (LH) sediment samples are shown.

Figure 4. (a) Organic carbon loadings for river sediments, and marine sediments. (b) OC versus MSA for soils samples within the basin. Symbol key and abbreviations are
as for Fig. 1 to 3. Blue shaded area corresponds to range typical for river suspended and non-deltaic sediments as described by Blair and Aller (2012).

Figure 5. Organic carbon loading versus (a) $\delta^{13}C_{OC}$, (b) $\Delta^{14}C_{OC}$, for Godavari river basin and marine sediment core 16A. $\Delta^{14}C$ values for marine sediments refer to the
age corrected value ($\Delta$). Symbol key and abbreviations are as for Fig. 1 to 3.

Figure 6. (a) LCFA loading versus distance to coast (b) $\delta^{13}C_{LCFA}$ versus distance to coast for river basin samples. Symbol key and abbreviations are as for Fig. 1 to 3.

Figure 7. Identifying major sources of organic carbon to the Godavari River and the offshore sediments core using stable ($\delta^{13}$C) and radio ($\Delta^{14}$C) isotopes (Modified after Marwick et al., 2015). The soil end-member is defined based on ranges of values observed within the Godavari and other tropical river systems (e.g. Pessenda et al., 1997; Shen et al., 2001; Trumbore et al., 1989). Symbol key and abbreviations are as 980     for Fig. 1 to 3.

Figure 8. Box-and-whisker summary of the geochemical and sedimentological data for the Godavari River basin and offshore sediments. (a) Organic carbon loadings, (b) mineral surface area (MSA), (c) bulk stable carbon isotope ratio, (d) long-chain fatty 985     acid loadings (e) mean weighted stable carbon isotope ratio of long-chain (C$_{26-32}$) fatty acid, and (f) bulk radiocarbon signature. The box represents the first (Q1) and third quartiles (Q3), and the line in the box indicates the median value. The whiskers extend to 1.5*(Q3-Q1) values, and outliers are shown as points. SS = surface soil; DS = deep soil; Rbe = riverbed; Rba = riverbank; EH = early Holocene; LH = late Holocene.

## Fig. 1

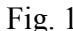

Fig. 2

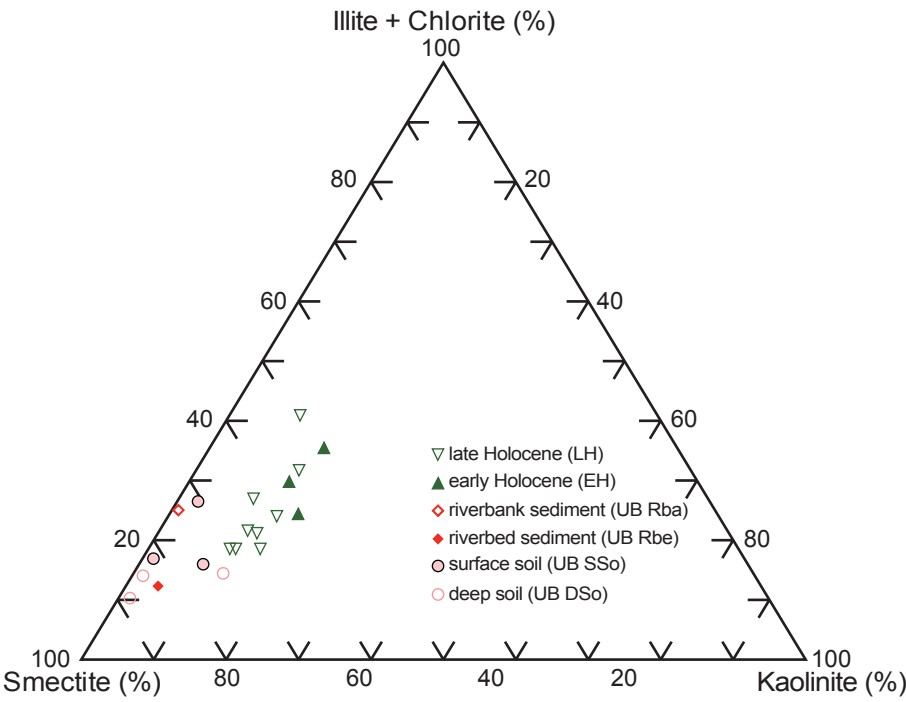

Fig. 3

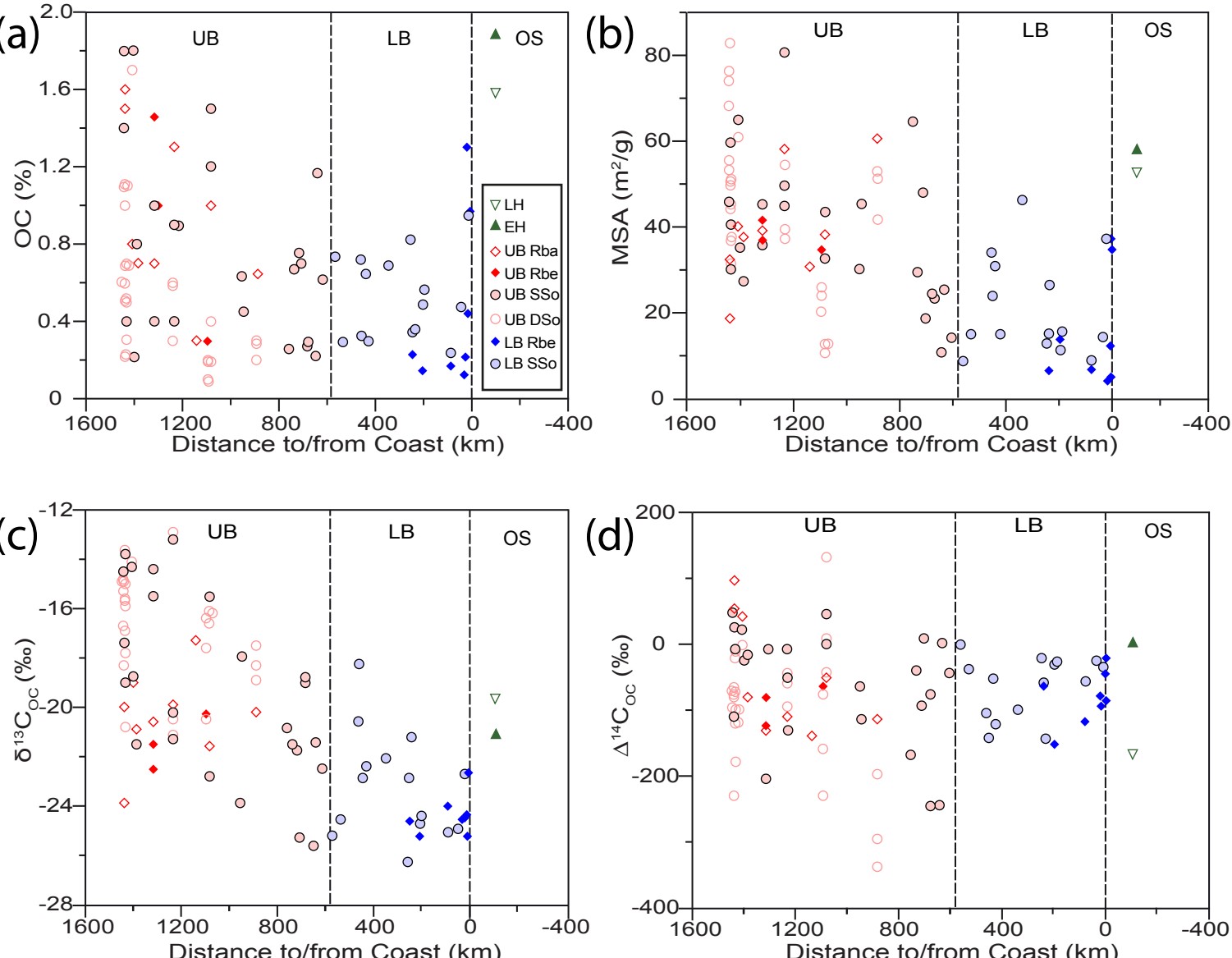

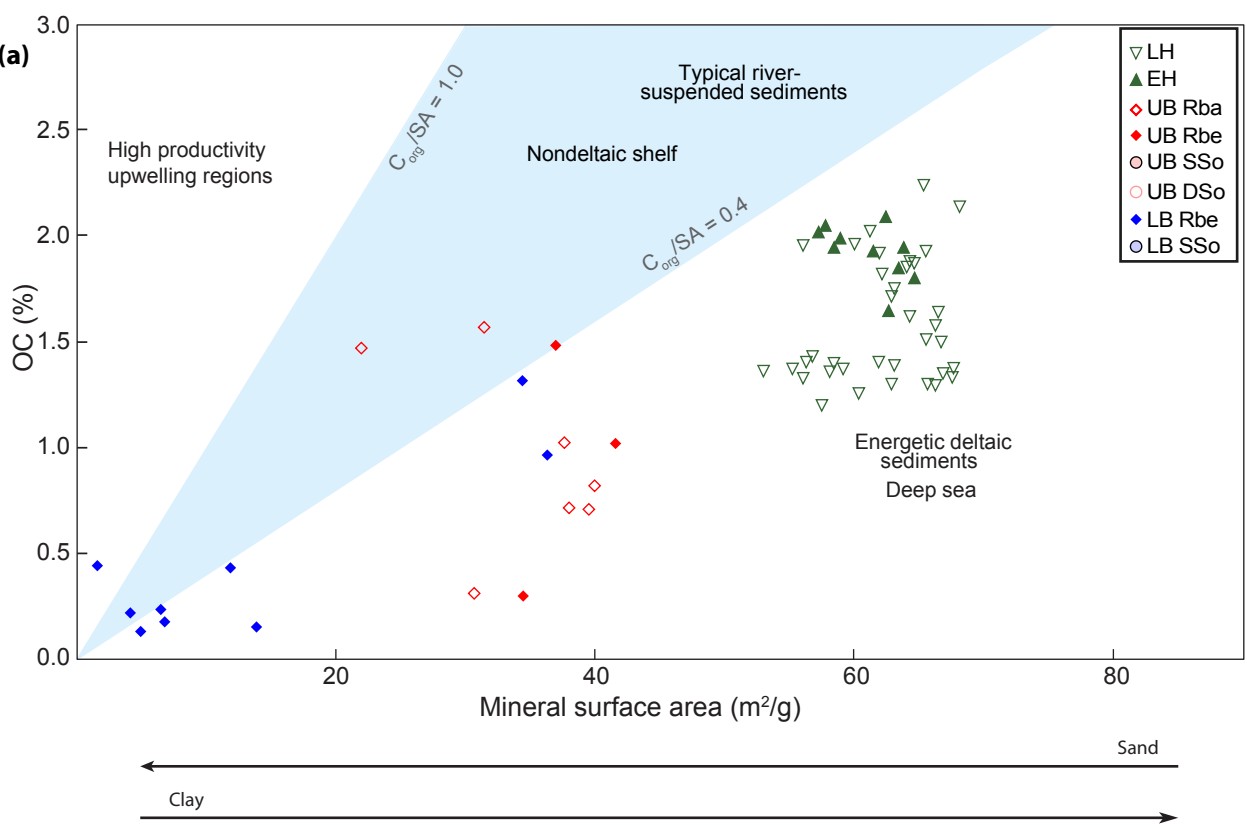

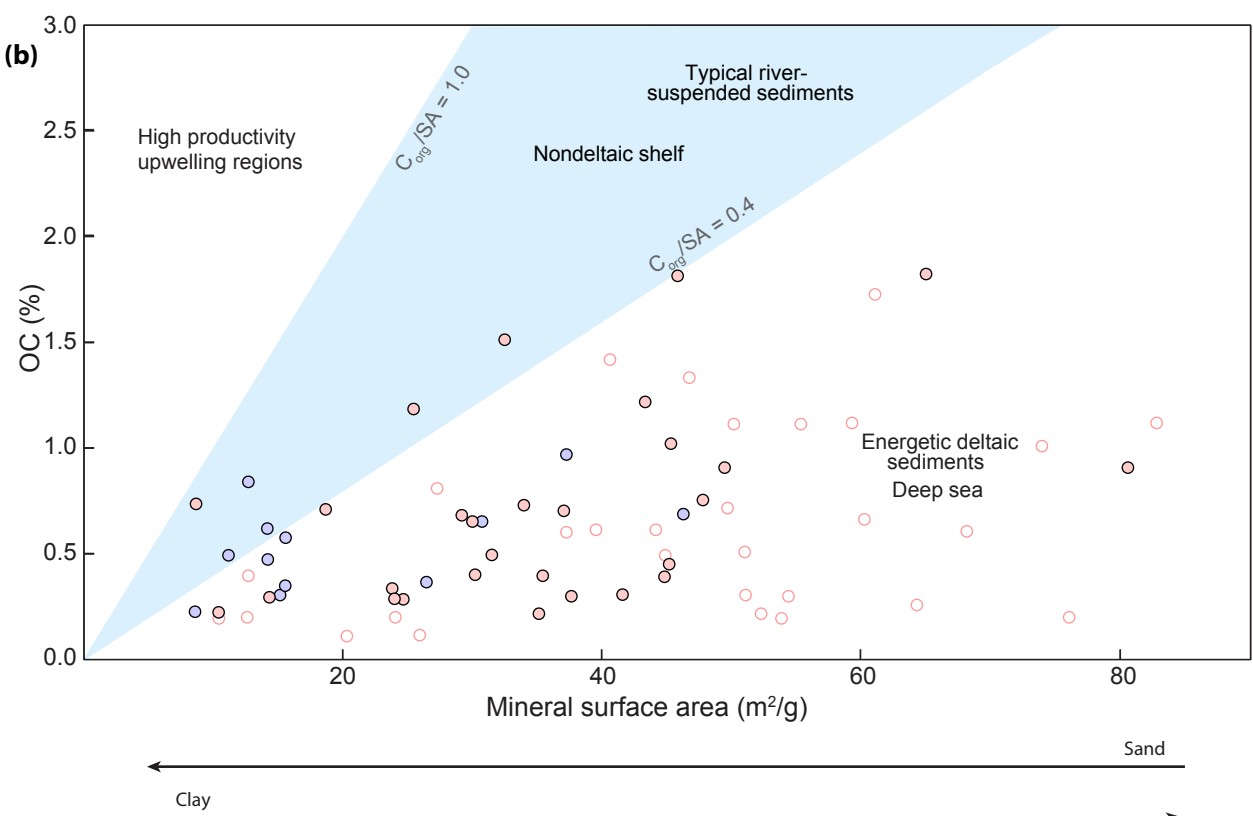

Fig. 5

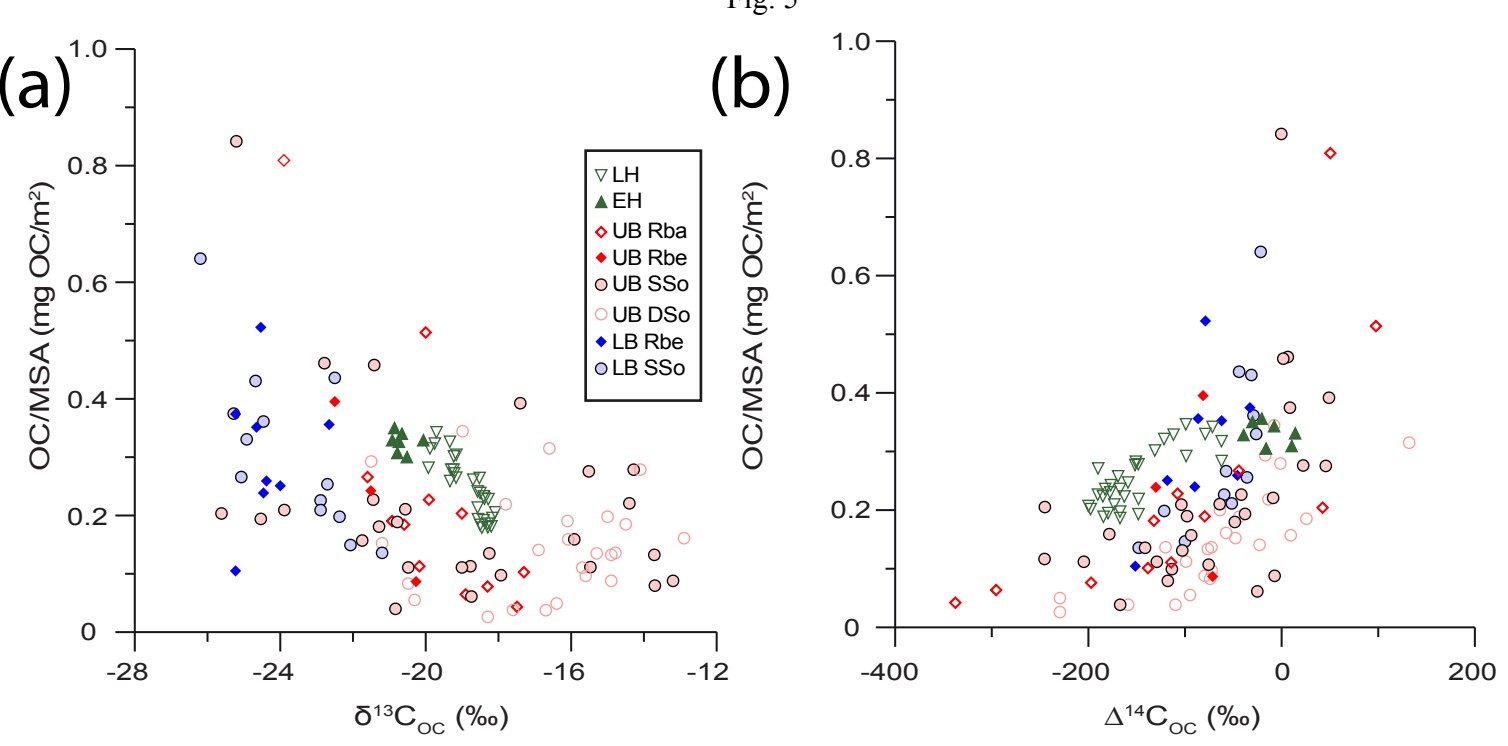

Fig. 6

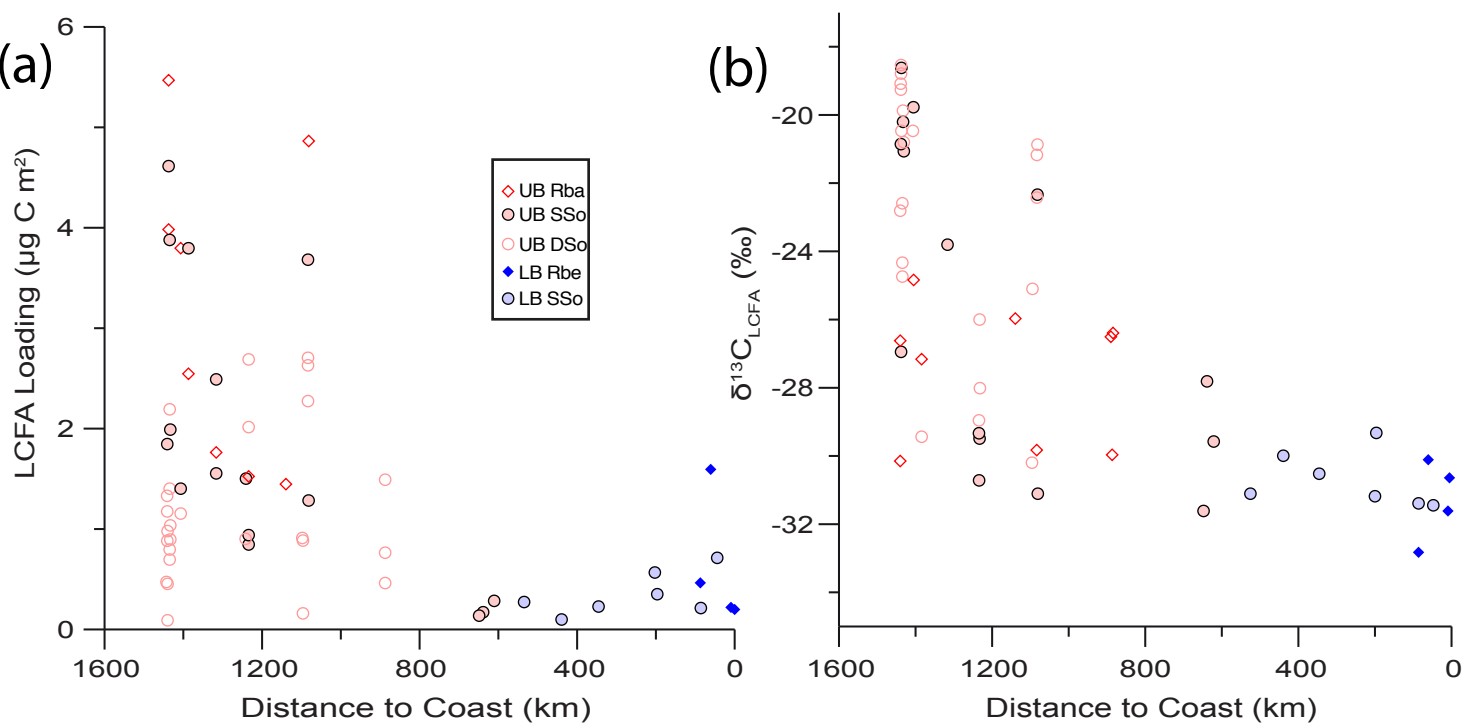

Fig. 7

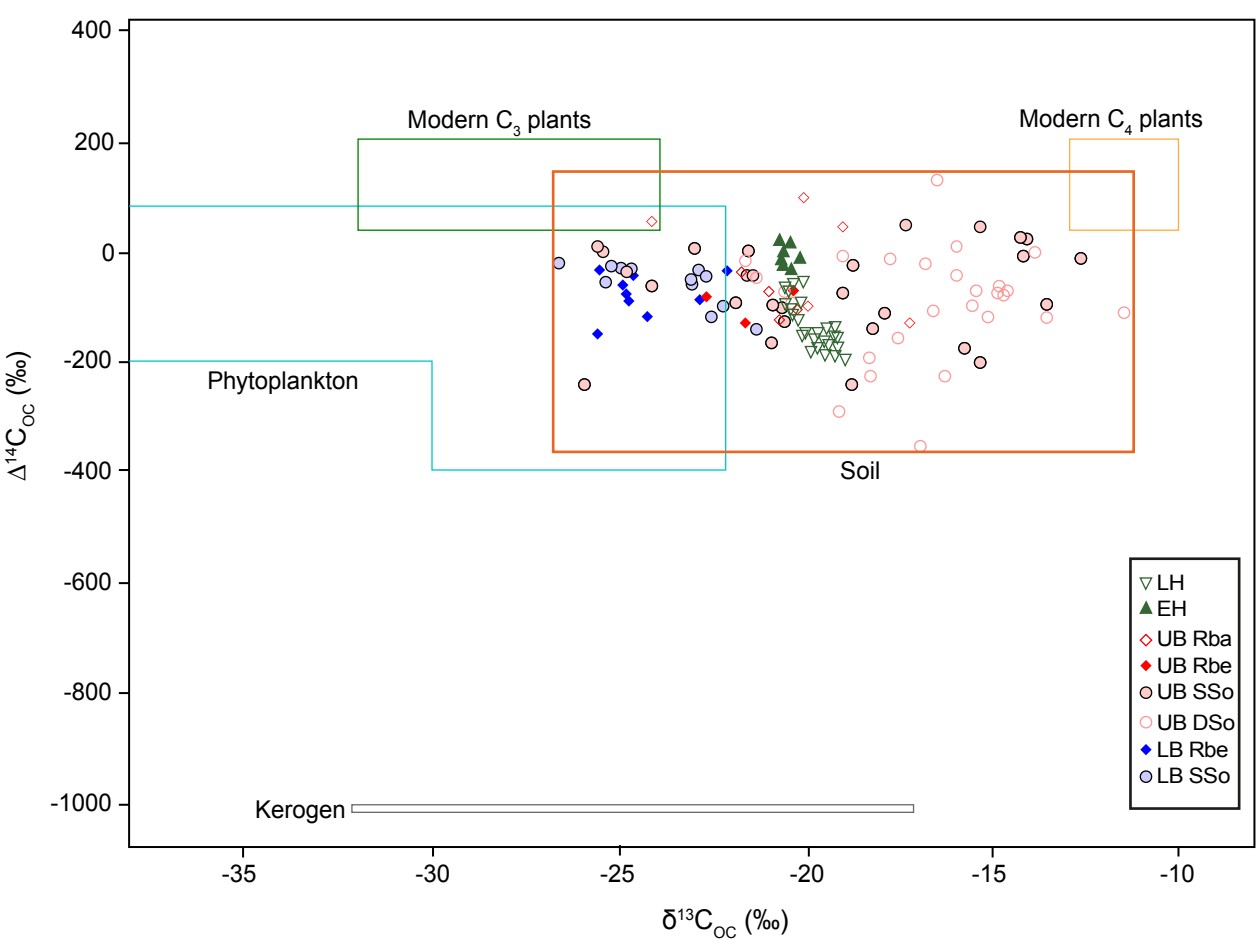

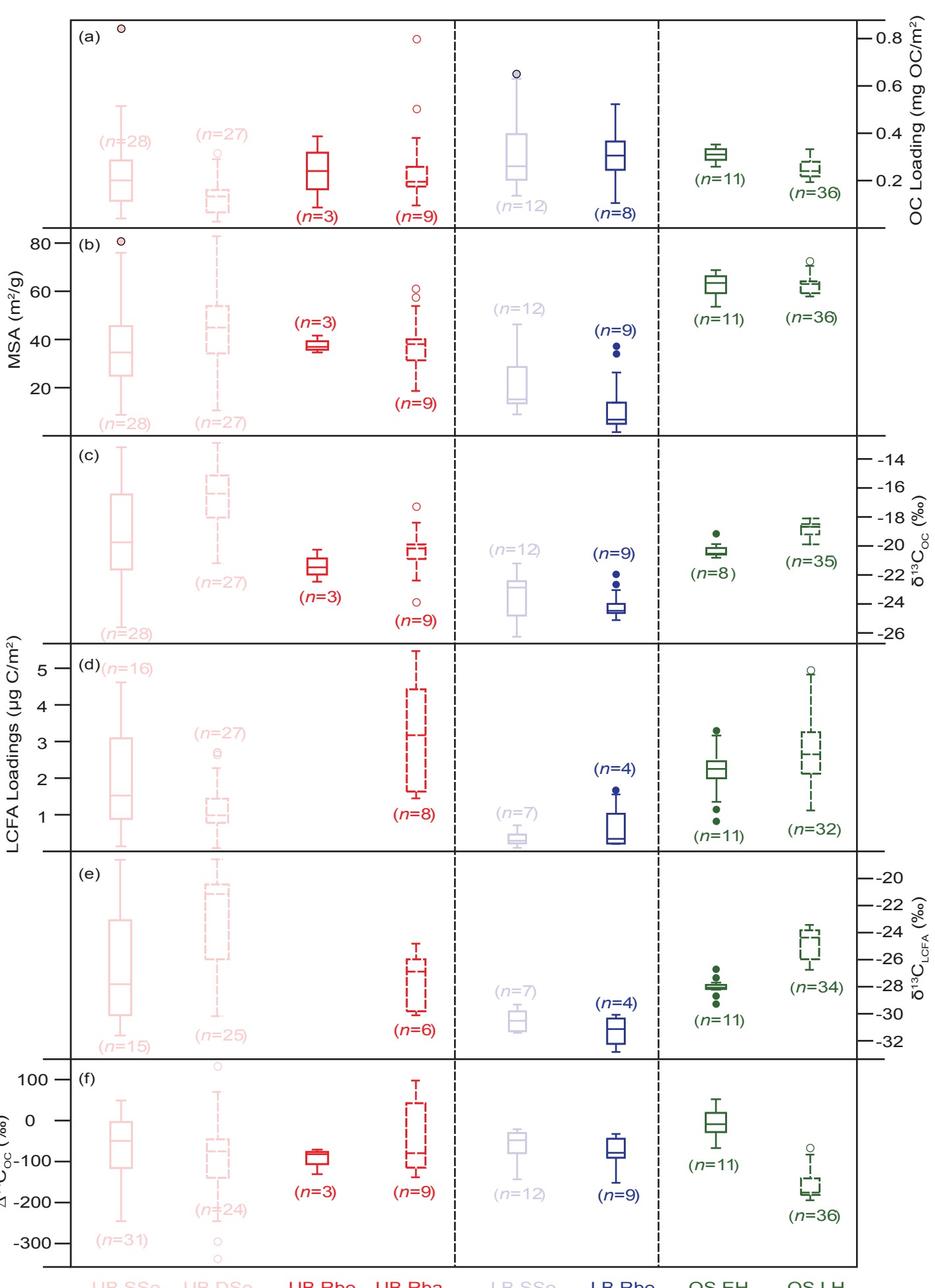