# Peer review of "Reconciling drainage and receiving basin signatures of the Godavari River system"

_Biogeosciences, 2018_

## Referee Comment (RC1) · Anonymous Referee #1 · 28 Mar 2018

First, several caveats— - I'm on extensive travel, and don't really have time to work my way through this quite dense manuscript. - I'm not a sedimentologist/mineralogist, so not well-qualified to speak to the technical details.

That said, I thought this manuscript was well-done. It laid out the issues clearly, worked through the problem setup and execution, and did a good job of working towards answering the two main questions posed.

---

## Referee Comment (RC2) · Anonymous Referee #2 · 25 Apr 2018

Review for Biogeosciences Discussions of Usman et al., Reconciling drainage and receiving basin signatures of the Godavari River system.

This paper examines the composition of organic carbon in river bank and river bed materials of the Godavari River. Recent studies (by some of the co-authors) have uncovered notable changes in the sedimentation rate, detrital provenance, and the bulk age of organic matter deposited offshore of this river system during the Holocene. This paper thus seeks to better understand the modern Godavari River system and the controls on the organic matter composition and its variability in the basin. The work forms part of a growing set of literature seeking to better quantify the composition and age of organic carbon in rivers.

The paper seeks to link the mineral surface area to the %OC, stable isotope composi-

tion and radiocarbon activity, and assess the 'loading' of long-chain fatty acids through-out the basin. Large contrasts are found between parts of the basin which are linked to vegetation, climate and lithology. The study then assesses how terrestrial soils and river sediments differ from the Holocene sediments offshore. Some of the results are very interesting (OC/MSA vs D14C and d13C for instance) and the discussion is very relevant to the readership at Biogeosciences.

I have a couple of points which require a little more clarity:

A) Nature of river sediments samples: I'm not sure what was sampled. The authors explain bank and river bed materials, and seem to suggest that these are representative of the materials sourced by erosion upstream. But what grain size range were captured by these samples? and do they represent the grain sizes carried by the river throughout its length? If they are sand-silt-clay, are these the only sediment grain sizes present on the river bed? there is no gravel? In short, what they actually tell you?

B) MSA control on %OC: There are a few points in the manuscript which talk about the importance of mineral surface area for organic carbon loading (e.g around line 520). But the data really don't support those statements (see Figure 4, where there are no links between these measurements and %OC is pretty similar where MSA varies much more). In contrast, the ratio of these measurements (OC/MSA) does appear to be linked to the 14C activity and stable isotope composition of the organic matter. Doesn't that suggest that the mineral surface control is linked to the residence time of organic matter and its processing in the catchments? Rather than its overall abundance in the sediments? This seems an apparent contradiction which is worth exploring.

To put it another way...

Figure 5 shows that generally, low OC/MSA is 14C-depleted (older). But low %OC is not necessarily associated with low MSA (e.g. ranging from 5 to 70 m2/g for %OC of ∼0.1%). Doesn't that suggest that the mineral surface area control is acting as protection (allowing organic matter to age), rather than promoting substrate for sorption

(line 523)?

Take the other side of the story, Figure 5 suggests high OC/MSA is younger. But high %OC is not necessarily associated with high MSA (e.g. ranging from 10 to 80 m2/g for %OC~-0.8%). So could this be due to an entirely different reason, decoupled from the surface area? Another explanation for this material would be that it represents organic matter not associated with any mineral – i.e. is discrete particles of organic matter, which may be more likely to be younger.

Some discussion on these points would be welcome.

Other comments with the line number:

30 – having read the paper, this final statement of the abstract seems to contradict those statements written on line 705 – does the abstract need some modification here?

255 – what grain size are the sediments on the river bed? What grain size was targeted? How do you know they are freshly deposited?

345 – do you mean DELTA14C? this could be clearer, as I think this is slightly different as it is being used to compare the marine sediment core measurements and modern.

420 – so are these river bed samples representative of the whole river material? If so this needs to be explained. Other studies have tended to focus on sampling the finer component of river flood deposits (sand and finer), as a way of linking (potentially) to the suspended sediments carried by the river. If this was the case here, then the grain size distributions could just reflect this sampling bias, and are not representative of parts of the basin as is suggested here.

520 – if the mineral surface area was important, why is the link between MSA and %OC so unclear when plotted? In figure 4 one could argue that MSA is not a control on %OC because there is no relationship across the sample set, nor in any one part of the sub samples.

540 - The bulk isotopic composition and radiocarbon activity are linked to the OC/MSA – this is interesting. The discussion was a little brief on this – particularly on the OC/MSA vs 14C link. Is this degradation signal, or a OC loading signal? Or both? And where is this happening, presumably in the soil sections? Or is there a role for floodplain processes or processing within the river corridors?

610 – "marked differences" – this wasn't so clear looking at the figures

Figures 2-5: the fact the stars are the marine sediment core samples could be much clearer. They don't have the same label as Figure 1 and the caption doesn't mention this explicitly. "SS" is used for suspended sediment in some studies so could be confusing.

Figure 5 and Figure 7: for the marine core samples, its unclear in the caption or figure whether these are the bulk 14C activity of OC, or a calculated D14C at time of deposition using the 14C-activity of the foraminfer (as per Giosan et al., 2017). Please clarify.

Figure 5 – the text mentions some of these soils samples are soil depth profile from a single location. It would be interesting to consider how these look in this space (i.e. to what degree is the signal in the river set by carbon cycling in soils?).

Figure 7 – the stars are hard to distinguish as filled (early Holocene) and open (late Holocene), please modify.

Figure 8 – the "soil" box doesn't seem to correspond to the soil samples?

---

## Author Comment (AC1) · 10 May 2018

We thank reviewer #1 for their comments and are pleased to know they find the manuscript acceptable as is.

---

## Author Comment (AC2) · 10 May 2018

The comment was uploaded in the form of a supplement:
https://www.biogeosciences-discuss.net/bg-2018-23/bg-2018-23-AC2-supplement.zip